# Improvement of Ruminal Neutral Detergent Fiber Degradability by Obtaining and Using Exogenous Fibrolytic Enzymes from White-Rot Fungi

**DOI:** 10.3390/ani12070843

**Published:** 2022-03-27

**Authors:** María Isabel Carrillo-Díaz, Luis Alberto Miranda-Romero, Griselda Chávez-Aguilar, José Luis Zepeda-Batista, Mónica González-Reyes, Arturo César García-Casillas, Deli Nazmín Tirado-González, Gustavo Tirado-Estrada

**Affiliations:** 1Facultad de Medicina Veterinaria y Zootecnia, Universidad de Colima, Tecomán 8930, Colima, Mexico; mcarrillo13@ucol.mx (M.I.C.-D.); jzepeda15@ucol.mx (J.L.Z.-B.); cesargarciacasillas@hotmail.com (A.C.G.-C.); 2Posgrado en Producción Animal, Departamento de Zootecnia, Universidad Autónoma Chapingo, Texcoco 56230, Edo. México, Mexico; microbiologia.pecuaria08@gmail.com; 3Centro Nacional de Investigación Disciplinaria Agricultura Familiar (CENID AF), Instituto Nacional de Investigaciones Forestales, Agrícolas y Pecuarias (INIFAP), Ojuelos de Jalisco 47540, Jalisco, Mexico; chavez.griselda@inifap.gob.mx; 4División de Estudios de Posgrado (DEPI), Tecnológico Nacional de México Aguascalientes (TecNM)/Instituto Tecnológico El Llano Aguascalientes (ITEL), El Llano 20330, Aguascalientes, Mexico; moni.rey3@yahoo.com.mx; 5Departamento de Ingenierías, Tecnológico Nacional de México Aguascalientes (TecNM)/Instituto Tecnológico El Llano Aguascalientes (ITEL), El Llano 20330, Aguascalientes, Mexico

**Keywords:** white-rot fungus enzymes, neutral detergent fiber, ruminal environment, animal productive behavior, fibrolytic enzymes immobilization

## Abstract

**Simple Summary:**

Neutral detergent fiber (NDF) quantifies the primary cell wall contents in ruminant feedstuff, and its degradability is related to intake and the quality of milk and meat production in ruminants. Overall, cellulases and xylanases, considered as endogenous fibrolytic enzymes (EFEs), can be obtained from white-rot fungi and utilized to increase ruminal NDF degradability. Although EFE can improve the NDF degradability and, therefore, animal productivity, results can be inconsistent. However, considering the ratio of cellulases to xylanases during the effect of EFEs on animal productive behavior has allowed us to find consistent correlations among previously published studies. The culture media where EFEs are obtained, in addition to the biotechnical tools—such as enzymatic activity analysis, metagenomics, metatranscriptomics, and enzyme immobilization—allow researchers to obtain or design products specific to ruminant feed applications.

**Abstract:**

The present review examines the factors and variables that should be considered to obtain, design, and evaluate EFEs that might enhance ruminal NDF degradability. Different combinations of words were introduced in Google Scholar, then scientific articles were examined and included if the reported factors and variables addressed the objective of this review. One-hundred-and-sixteen articles were included. The fungal strains and culture media used to grow white-rot fungi induced the production of specific isoforms of cellulases and xylanases; therefore, EFE products for ruminant feed applications should be obtained in cultures that include the high-fibrous forages used in the diets of those animals. Additionally, the temperature, pH, osmolarity conditions, and EFE synergisms and interactions with ruminal microbiota and endogenous fibrolytic enzymes should be considered. More consistent results have been observed in studies that correlate the cellulase-to-xylanase ratio with ruminant productive behavior. EFE protection (immobilization) allows researchers to obtain enzymatic products that may act under ruminal pH and temperature conditions. It is possible to generate multi-enzyme cocktails that act at different times, re-associate enzymes, and simulate natural protective structures such as cellulosomes. Some EFEs could consistently improve ruminal NDF degradability if we consider fungal cultures and ruminal environmental conditions variables, and include biotechnological tools that might be useful to design novel enzymatic products.

## 1. Introduction

Forage’s cell walls are composed of different proportions of cellulose, hemicellulose, pectin, lignin, and minerals that vary among the species and growth stages of plants [1]. Neutral detergent fiber (NDF) quantifies most cell walls’ components [2]. Cellulose and hemicellulose are the most abundant constituents of cell walls, but their energy availability for ruminants depends on the NDF composition and proportion [3,4,5], and on the efficiency of ruminal fibrolityc enzymes and bacteria in digesting those components [6].

In arid and semi-arid areas, forages, scrubs, straws, and stover are included in ruminant diets; however, their high content and structure of NDF might reduce the dry matter (DM) intake, with negative effects on animal production [7,8,9,10,11]. The high amounts of lignified components and the hydrolysis-resistance of the crystalline structures reduce the availability of the energy contained in the NDF [12].

Exogenous fibrolytic enzymes (EFE) are produced by ligninolytic fungi and have been used in ruminant diets to improve the availability of sugars from the cell wall [13]. The removal of lignin and/or hemicellulose is the first step to exposing the cellulose to an attack by cellulolytic enzymes in order to release glucose, due to the hydrolysis of the polysaccharide fraction of NDF to fermentable sugars. Although, the effects of including cellulases and xylanases on ruminant performance are highly variable, due to many external factors [13,14,15,16,17], some enzyme mixtures and doses, under particular ruminal conditions, can promote optimal hydrolysis of cell walls’ components to release sugars; moreover, they can improve milk yield and quality, as well as average daily gain (ADG) [14,15,16,17,18]. 

The application of biotechnology tools could help to design and improve enzyme extracts and enhance their activities in a ruminal environment [19]. This review examines the strategies for obtaining EFE products from white-rot fungi and enhancing their effect on NDF ruminal degradability.

## 2. Natural Production of Multi-Enzyme Cocktails by White-Rot Fungi

White rot fungi produce EFEs to adapt and grow in lignocellulosic substrates. Obodai et al. [20] found that *Pleurotus ostreatus* growth is positively correlated with the initial cellulose content (R^2^ = 0.60), hemicellulose (R^2^ = 0.62), ADL (R^2^ = 0.70) and crude fiber (R^2^ = 0.70), and *Trichoderma viride*, *Paecilomyces farinosus*, *Wardomyces inflatus,* and *P. ostreatus* produce fibrolytic enzymes when they are cultivated in microcrystalline cellulose [21]. 

Extracellular enzymes are produced by a specific gene expression induced by the culture media [22]. Gene expression can also be modulated by the amount of nitrogen compounds [23], ions [24], and the media state (solid or liquid) [25]. According to the culture media, some fungi produce isoenzymes that show stability at certain temperatures and pH levels, and an affinity to active sites, isoelectric points, molecular weights, specificity, spectral characteristics, and contents of sugars [22,26,27,28,29]. With the wide variety of forms and isoforms of cellulases and xylanases produced by fungi, it is possible to explore conditions that permit the production of enzymes that best fit certain biotechnological processes [28,29,30]. 

To access cellulose and hemicellulose energy availability, in nature, fungi require the coordinated action of ligninases, cellulases, and xylanases acting synergistically [26,27,30]; therefore, lignocellulose hydrolysis also requires the coordinated action of cocktails of enzymes [31,32]. Tirado-González et al. [30], found that NDF, acid detergent fiber (ADF), and ADL culture medias extracted from different corn stover hybrids promote different production of exogenous cellulolytic and xylanolytic enzymes ratios of *Bjerkandera adusta*, *Sporotrichum pulverulentum*, *Trametes trogii*, and *P. ostreatus*; thus, culture media can induce specific forms and isoforms of enzymes that are useful in degrading the cell walls of the high-fibrous forages included in ruminant nutrition.

Plant cell components have two fractions, the cellular contents (non-fibrous carbohydrates (NFCs)), and the cell wall’s structural carbohydrates (NDFs). The NFCs are a non-lignified material that is soluble in neutral detergent, and are composed of lipids, sugars, organic acids, non-protein nitrogen, soluble protein, and pectin, with high energy availability. On the other hand, NDF is primarily composed of lignocellulose, a hardly hydrolysable rigid structure formed by polymers of cellulose, hemicellulose, and lignin. Lignocellulose constitutes 30% to 70% of the DM of forages, and are also the major reservoirs of carbon fixed in nature [1,3,4,5].

Cell walls have three distinct morphological layers: (1) the primary wall, composed of 9% to 25% cellulose, 20% to 50% hemicellulose, 10% to 35% pectic substances, and 10% proteins; (2) pectic substances which stick adjacent cells together to form a middle layer; and (3) the secondary wall, consisting of mature xylem tissues that support the plant, composed of 41% to 45% cellulose, 30% hemicellulose, and 22% to 28% lignin [33,34]. 

Cellulose makes up to about 50% of the planet’s biomass. Its molecule is composed of glucose monomers (50 to 15,000) linked by β-glycosidic bonds, and its chains are arranged in a highly dense fiber package from parallel linear chains linked by electrostatic attraction and Van der Waals forces [34]. Cellulose surrounds the cell with the amorphous heteropolymer matrix of hemicellulose [35], primarily composed of pentoses, hexoses, and uronic acid [33,34]. Cellulose and hemicelluloses are synthesized at a similar time. 

Hemicelluloses are polysaccharides that usually have an equatorial β-1,4-linked glycosyl residue backbone (similar backbone linkages as the glucan chains that make up crystalline cellulose). Hemicellulose consists of a heterogeneous group of polysaccharides, including D-xylose, D-mannose, D-galactose, L-arabinose, D-galactose, and 4-O-methyl-D-glucuronic acid, and small amounts of L-rhamnose and L-fucose units [33,34,36,37]. According to their molecular weight, the properties of hemicelluloses and cellulose impact their potential industrial, pharmaceutical, and cosmetic applications; however, like cellulose, the hydrolysis of hemicellulose is a source of energy in ruminal fermentation [38]. 

In maize stover, galactose and arabinose begin to accumulate early in cell wall development, with primary wall growth occurring during internode elongation, and the major secondary wall constituents (glucose and xylose) accumulate rapidly until shortly before internode elongation ends [39]. Overall, xylans (homoxylans and heteroxylans, β-1,4-linked xylan backbone) are the predominant hemicellulose polysaccharides in hardwoods and herbaceous biomass. Arabinoxylans (arabinose as the primary side group attached at position 3 of xylose units in the backbone structure) are the major hemicellulose polysaccharides of straw and stover biomass; however, glucuronoxylans, mannans, homomannans, galactomannans and galactoglucomannans are the most abundant in hardwoods and softwoods. Plant galactans are minor hemicellulose polysaccharides and can be divided into sulfated galactans and arabinogalactans.

Hemicellulose is associated with cellulose microfibers [35] and is joint to lignin, a recalcitrant hydrophobic amorphous polymer embedded in cellulose and hemicellulose. In the laboratory, the quantification of lignin can be performed through hydrolysis using permanganate (lignin fraction of manganate) or the application of 72% H_2_SO_4_ to solubilize cellulose (lignin fraction in acid detergent (ADL)) [2]. Lignin is composed of cinnamyl alcohols derived from: *p*-coumaryl (*p*-hydroxyphenyl or H units), coniferyl (guaiacyl or G units) and sinapyl (syringyl units or S). In the polymerization, the precursors are oxidized by peroxidases to phenoxy radicals, and they react randomly and cause multiple lignin structures. In grasses and legumes, lignin formation comes from the rate of change of S to G units.

Lignin biosynthesis in grasses begins with the deamination of l-phenylalanine or tyrosine by ammonia lyases, yielding cinnamic or *p*-coumaric acids, respectively. The successive steps of hydroxylation and methylation lead, at first, to the formation of hydroxycinnamoyl-CoA thioesters, then hydroxycinnamaldehydes, and finally, *p*-hydroxycinnamyl alcohols (monolignols: *p*-coumaryl, coniferyl, and sinapyl alcohols); these undergo dehydrogenative polymerization via oxidases to form lignins, mainly guaiacyl, (from coniferyl alcohol) and syringyl (from sinapyl alcohol) [40]. The unions between lignin components could be of a condensed type (CC bonds) or of an uncondensed type (aryl alkyl ether linkage) of aromatic rings and propylic chains. The CC bonds determine the high condensation of lignin, and they are only formed between H or G units [3,33,34]. 

Lignin is thought to be associated with hemicelluloses through covalent and non-covalent bonding to cell walls’ carbohydrates (LC), although there are different types of LC linkages such as benzyl ether, benzyl ester, glycosidic or phenyl glycosidic, hemiacetal or acetal linkages, and ferulate or diferulate esters [41,42]. However, ferulate and diferulate esters are the most abundant LC linkages in grasses, as the presence of carboxylic acid groups at the end of their propenyl groups produce ester linkage with the polysaccharides [42]. 

In grasses, hydroxycinnamic acids, *p*-coumaric, and ferulic acids are ester- and ether-linked to cell wall polymers [40], while sinapic acid esters have been implicated in cell wall cross linking. During lignification, ferulate and diferulate copolymerize more and ferulate monomers are linked to lignins by various types of ether and CC bonds. At the same time, *p*-coumarate increases [40]. Ferulic acid can oxidatively crosslink to form intermolecular ester-bonds with another arabinoxylan, and ester–ether bonds between polysaccharide and lignin (arabinoxylan–ferulate–lignin). Diferulic acids have mainly been detected in the high-arabinose substitution region of arabinoxylan [28]. Then, arabinoxylans of hemicellulose can be covalently cross-linked with lignin through ferulic acid and exported to the maturing wall. Since S units have two methoxy groups at positions 3 and 5 and do not establish unions with other lignin components’ CC bonds, xylans can bind to lignin during quick polymerization of lignin monolignols [3]. 

Cross-linking of cell wall polysaccharides and lignin by hydroxycinnamic acids increases the strength of the plant cell wall, decelerates wall extension, and acts as a barrier to block the ingress of microbial invaders as well as hydrolytic enzymes [28].

Lignin hydrolysis is complex and, therefore, primarily limits access to plant NDF and NFC [43]. The degradability of cell walls is mainly related to the structural changes that occur in the cell wall polymers according to the type and stage of maturity of the plants [1,3,4,34,43,44], and to a lesser extent, its concentration [7]. 

Because of their cell walls’ composition and structures, grasses are more digestible than legumes [3]. The cell walls of legumes have more cellulose and xylan content, while grasses have few pectins and more ferulate contents [1,45], and C4 grasses tend to have higher hydroxycinnamic acids than C3 grasses [40]. 

In maize stover, the ferulate-to-cell-wall degradability ratio is negative during elongation and positive as it passes maturation time, until senescence initiates, when it becomes negative again [40]. Jung and Casler [1,45] found that for total ester and ether ferulates, the siringil-to-guaiacyl ratio, *p*-coumarate, and the arabinose-to-xylan ratio increased, while for total ferulates, arabinose ratios decreased across the maturation time of two maize hybrids; accordingly, the 24 and 96 h in vitro degradability of glucose, galactose, xylose, arabinose, mannose, and uronic acids reduced within that time. 

During the elongation of plants, diferulates’ ester bonds are crossed with the arabinoxylan to join the middle lamella hemicellulose with the secondary wall; meanwhile in the maturation stage, the secondary wall is thickened by an increased content of coumarate and the ether bonds of ferulate. After aging, the proportion of the secondary wall ferulate decreases by increasing other phenolic compounds. This results in a lower proportion of ferulate to the rest of the phenols present, which makes the DM less digestible [1,3,4,45]. 

Phenol oxidases (Laccases (Lac, E.C. 1.10.3.2), manganese peroxidases (MnP, E.C. 1.11.1.13) and lignin peroxidases (LiP, E.C. 1.11.1.14), etc.) are firstly produced by white-rot fungi [46], allowing them to depolymerize, repolymerize, dimethylate and oxidize phenolic and non-phenolic compounds, aniline and xenobiotics [47]. These enzymes recognize catalytic domains, where they bind to the cell wall to degrade the ADL [48]. The NDF component degradation by fungi begins with the action of low-molecular-weight mediators such as O_2_, H_2_O_2_ and Mn^2+^ [49,50]; then, the Lac mediator system acts in the ligninases, coupling the reduction of a dioxygen electron to two water molecules [51]. This oxidation reaction releases free radicals that act as intermediaries between substrates and enzymes. The four copper ions (Cu^2+^) of the catalytic oxide-reduction (redox) center of Lac act through mediators with high redox potential that also allow non-enzymatic oxidative polymerization or depolymerization routes [50,52,53].

After that, LiP oxidizes substrates through several steps of electron transference and creates intermediary radicals as phenoxy [54]. This leads to a large number of non-enzymatic polymerization reactions and rearrangements of dimethylation and intramolecular addition, allowing the oxidation of many non-phenolic aromatic substrates that do not require mediators due to their redox potential. Finally, manganese peroxidase—which contains a heme molecule as protoporphyrin iron—catalyzes the dependent oxidation of Mn^2+^, which is then left on the surface of the enzyme in the complex with oxalate and other chelates, wherein the Mn^3+^ acts as a redox mediator whose potential is limited only by the phenolic lignin structures. Organic acids such as oxalate and malonate, are primary compounds that act as second mediators in the production of reactive free radicals. Examples of this are: the carbon radical center (acetic acid), peroxy radical superoxide (O_2_), and radical (CO_2_) in the absence of H_2_O_2_; these radicals can be used by MnP to increase the efficiency of lignin degradation [52,53].

The fungal members produce carbohydrate esterases, such as acetyl xylan esterases (AXE, E.C. 3.1.1.72) and feruloyl esterases (FAE, E.C. 3.1.1.73), that are essential to catalyzing the hydrolysis of ester linkages between acetyl groups and xylan (AXE), and the ester linkages between hydroxycinnamic acids from xylan and pectin (FAE) [29]. FAEs act as accessory enzymes to assist xylanolytic and pectinolytic enzymes in gaining access to their site of action during biomass conversion; they are responsible for removing ferulic acid residues and cross-links from polysaccharides [28]. Previously, phylogenetic analysis classified the FAE as primarily obtained from *Aspergillus* sp., *Pichia pastoris, Neurospora crassa, Chaetomium* spp., and *Talaromyces funiculosus*.

Additionally, white-rot fungi produce cellulolytic and xylanolytic polysaccharidases. The cellulolytic complex constitutes the endo-β-glucanases (endo-[1, 4]-β-D-glucanase, E.C. 3.2.1.4); exo-β-glucanases (exo-[1, 4]-β-D-glucanase or cellobiohydrolase, E.C. 3.2.1.91); and β-glucosidases (cellobiases, E.C. 3.2.1.21) [55]. These enzymes work as follows: First, the endo-β-glucanase randomly breaks β-glycosidic internal links in amorphous regions of the cellulose molecules, promoting a rapid decrease in chain length and a slow increase of free reducing groups, thus allowing the formation of attack sites for exoglucanases [20]. After that, exo-β-glucanase gradually attacks the non-reducing ends of cellulose molecules, releasing cellobiose subunits, which results in a rapid increase in sugars or reducing groups but little change in the size of the polymer. Finally, the β-glucosidase hydrolyzes both cellobiose, produced in the previous steps, and low-molecular-weight cellodextrins [20]. Enzymes that comprise the hemicellulolytic complex hydrolyze arabinose side chains (arabinofurosidase, E.C. 3.2.1.55), release acetate groups (acetyl xylan esterase, E.C. 3.1.1.72) and remove the side chains of glucuronic acid from xylose units (glucuronidase, EC 3.2.1.131). Additionally, they act on short oligosaccharides to produce xylose (β-1,4 bonds hydrolyzing aryl xylopyranoside; β-xylosidase, 3.2.1.37) on xylans and xylo-oligosaccharides, to produce a mixture of xylooligosaccharides (endo-β-xylanase (endo-[1, 4]-xylanohydrolase or D-xylanase, E.C. 3.2.1.8)) [56].

As in nature, efficient enzyme cocktails are required to complete the hydrolysis of lignocellulosic biomass. For example, in increasing the ratio of endoglucanases, xylanases could improve the NDF degradability of alfalfa and corn silages [57]. Fortes et al. [58] tested fungal enzyme extracts on the degradation of the lignocellulosic complex sugarcane. They found that the cellulases and xylanases of *Trichoderma reesei* and *Aspergillus awamori* were acid ferulic esterases (1700 vs. 420 UI/L); endoglucanases (CMCase; 20,000 vs. 4900 UI/L); β-glucosidase (340 vs. 45,600 UI/L); and xylanases (12,600 vs. 79,000 UI/L), respectively, and that the enzymatic activity of *T*. *reesei* and *A. awamori* enzyme mixtures acted synergistically, 2-fold. Mkabayi et al. [32], tested the synergistic action of two feruloyl esterases (FAE5 and FAE6) and an endo-xylanase (Xyn11) from *Thermomyces lanuginosus*; they were applied to 1% (*w/v*) corn cobs. The combination of 66% Xyn11 and 33% FAE6 displayed an improvement in reducing sugars compared to Xyn11 alone. Xylanases generate ferulated xylo-oligosaccharides (XOS), which become substrates for FAE, liberating ferulic acid as a product. The removal of ferulic acid then increases the accessibility of xylanases to XOS, for further hydrolysis into shorter XOS and/or xylose. 

## 3. Exogenous Fibrolytic Enzymes in Ruminant Diets

Some authors have hypothesized that NDF proportion and degradability should be the most important predictors of animal productive behavior [11,59]. NDF might be incorporated according to the partial rate and total degradability, since an excess of it should reduce the passage rate and, therefore, the DM intake, potential milk, and meat production [7,8,9,10,60]; these justify the study of many alternatives to increase the potential degradability of NDF. Cellulase and xylanase biotechnology began in the early 1980s, in the study of the action of enzymes extracted mainly from *Aspergillus* sp. and *Trichoderma* sp. as treatments to improve the quality of feed for animals [61].

Neylon and Kung [62] reported a positive relationships between the increased in vitro NDF degradability (IVNDFD) of hybrid sorghum, and maize and milk production. Oba and Allen [7] found that cows fed exclusively with 56% IVNDFD of forages produced an equal amount of milk to those fed a diet that included 30% grain and forage with lower digestibility (IVNDFD = 46.5%). EFEs might be useful to (1) increase NDF digestibility (NDFD); (2) increase the digestible energy intake; (3) reduce feed conversion (FC) in milk and meat; and (4) decrease production costs [14]. Some cellulase and xylanase enzyme extracts can improve NDFD, as well as meat and milk production [16]. The use of cellulases [63], xylanases [64], ferulic acid esterases [65], and combinations of cellulase and xylanase [63] may improve the in vitro digestibility of DM (IVDMD) and IVNDFD. However, the effects of EFEs are primarily related to factors such as the cellulase-to-xylanase ratio; the type and doses of EFE; the types of ruminants, plants (legumes or grasses), or studies (in vivo, in situ, or in vitro); the ingredients of the diets; and the forage-to-concentrate ratios [16,17,30,63].

Many EFE products are evaluated with ruminal liquid through in vitro experiments, considering that in vitro experiments can be easily standardized; however, IVNDFD has been related with passage rate, the feeling of fullness (satiety) in the rumen, DM intake, and milk production [7,66]. 

In vitro results of EFE addition are not always relatable to in vivo effects, but sometimes they are comparable; for example, Eun et al. [57] found that some doses of endoglucanases and xylanases could enhance the IVNDFD; this is accordance with in vivo studies that show an increase in NDFD when EFEs are added to ruminant diets [67]. However, the in vitro studies of the effect of EFE addition to ruminant diets allow us to understand the interactions among the types of activities of EFEs and the ruminal liquid environment, with more consistent results than in vivo studies [57,68,69].

In dairy cattle, EFEs can improve DM intake, gross energy intake, production, and milk composition in lactating cows [70,71]. Regarding beef cattle, Beauchemin et al. [13] found that as a result of the application of an enzyme product to a diet containing 95% barley grain, feed efficiency improved from 6% to 12% depending on the dose of enzyme used. The positive effects on animal productive behavior could be related to increased digestibility of NDF and ADF in the diet when EFEs are included [72]; for example, Krause et al. [73] observed a 28% increase in ADF digestibility upon inclusion of exogenous enzymes in the diet. However, sometimes, adding EFEs does not have positive effects on DMD, NDFD [74] or DM intake [70]; milk production [13]; or ADG. Consistent results have been documented in meta-analyses, such as the improvement of milk production and composition; FC; and ADG [15,16,17].

## 4. Enhancing Ruminal NDF Degradability through Fibrolytic Enzymes

The combination of different enzymatic activities could have positive effects on fiber degradation [6], but different fungus-combined extracts may have synergistic or negative interactions [26]. Sufficient supplementation of cellulase activity in relation to xylanases (cellulase-to-xylanase ratio) is the key to determining the optimum dose of enzyme extract for fiber degradability. Although, it is difficult to predict the potential of enzyme extract in the rumen as a reference of its enzymatic activity, authors have agreed that results could be more consistent between the evaluations without rumen fluid of the same extract, considering the relationship between cellulases and xylanases, enzyme stability and structure, depending on the fungi from which the extract was obtained [16,26,30]. In many cases, the products evaluated in ruminants do not contain an appropriate combination of types of enzymes, which may reduce their effect on fiber digestibility [57,75].

Since fungal gene expression to produce forms and isoforms of enzymes can be modulated by culture media [23,24,25], the first step to obtain EFE products with ruminal feed applications is the selection of specific culture media to obtain optimal cellulase-to-xylanase ratios, to improve the degradability of NDF of forages [30].

Cellulase-to-xylanase ratios from 1:100 to 1.5:1 have improved the degradability of NDF alfalfa and corn silage (evaluated in vitro with rumen fluid). They were linearly associated with the total dose of cellulase supplemented (R^2^ = 0.58 to 0.75) [76]. Depending on the type of forage, this could indicate a greater relationship between the improvement of the degradability of NDF and the activity of certain types of cellulases, such as endoglucanases, exoglucanases, β-glucosidases, or a combination of cellulase and xylanase activity. Eun and Beauchemin [76] explain that the use of xylanases in proper combination with cellulases improves glucose release by increasing xylose removal. It is possible that by removing hemicellulose, some hemicellulose–cellulose bonds are broken, increasing the accessibility to cellulose. Consequently, pre-treatments such as washing, the use of ligninolytic enzymes, polyethylene glycol (PEG), and surfactants, could improve the effect of cellulase and xylanase enzymes on NDFD [77].

In recent years, research has been conducted regarding the use of cellulosomes for more efficient use of lignocellulose. Resch et al. [78] examined a cocktail of three enzymes of the fungus *Hypocorea jecorina* and the cellulosome of *Clostridium thermocellum*. These two combined systems have a better effect on the degradation of microcrystalline cellulose, which can help to design products that could degrade certain kinds of recalcitrant polysaccharides.

Synergy and interactions among cell wall composition, ruminal microorganisms, and endogenous fibrolytic enzymes should be considered when analyzing the potential effects of EFEs [13,79,80,81]. Biotechnology tools like metagenomics and meta-transcriptomics might allow for a better understanding of the effects on the ruminal environment when EFEs are added to ruminant diets [79,80,81,82,83]. Changes in the ruminal environment affect volatile fatty acids (VFA), fatty acids biohydrogenation and, therefore, milk and meat production, quality, and shelf-life [84,85,86]. Additionally, to obtain EFE products to improve NDFD, ruminant-specific conditions should be considered.

### 4.1. pH and Temperature

The optimal ruminal pH usually ranges from 6 to 7. Feed fermentation in the rumen produces VFA and, occasionally, lactic acid. Different types of ruminants, physiological animal stage, diet, fermentation patterns, and environmental temperature promote ruminal pH and temperature changes. High concentrate/grain diets reduce the ruminal pH [35,44]. Low ruminal pH has a negative effect on ruminal fermentation and microbial growth [69,87], inhibiting fibrolytic microbiota growth and reducing fiber degradability [88]. Therefore, a reduction in pH may have negative effects on the availability of energy from NDF components [89]. The degradation of cellulose increases when the pH is over 6.0 [90]. The average ruminal temperature ranges between 38 and 42 °C [91]. Optimal fermentation occurs after feed intake, and rumen temperature reaches 41 °C [92]. Experiments carried out with ruminants, have shown a negative correlation between ruminal pH and the temperature of the rumen [93,94] and, also related to high-grain diets, EFE supplementation can improve NDF when ruminal pH is under 6.0. Fungi’s endoglucanases, exoglucanases, and β-glucosidases and ligninases act optimally when pH ranges from 4.5 to 5.5 [95]. Since EFEs’ optimal activity is shown to occur at over 50 °C [83], they might be highly stable in the rumen, even in ruminants fed high-grain diets.

### 4.2. Osmolarity

Normal osmolarity in the rumen ranges from 240 to 400 mOsm/L [96,97]. Variations in osmolarity depend on energy and DM intake, since there is a release of cations and anions (such as phosphates, sulfur, and chlorine) from the digested diet’s compounds. High-grain diets enhance acute acidosis in ruminants (pH ≤ 5.0), derived from higher levels of ruminal osmolarity [98]. A highly acidic rumen environment increases bloodstream acidosis when AGV’s are not absorbed at the rate at which they are produced. High osmotic pressure in the rumen causes water to be quickly pulled inward into the blood through the rumen wall, to neutralize the osmotic pressure and to increase the ruminal pH through the removal of hydrogen ions [98,99]. Fungal EFEs are stable in a ruminal osmolarity environment. Sun et al. [100] reported the activity of cellulases produced by *Trametes ressei* using rice straw as a treated substrate in alkaline conditions for biofuel production. Research has even been conducted with some cellulases called halophilic cellulases, which work well in environments with high osmotic pressure [101]. It is feasible that the osmotic pressure of the rumen enables the activity of enzymes that degrade the fibers of the lignocellulosic materials.

### 4.3. Proteolityc Activity

Important proportions of protein are degraded at the beginning of ruminal fermentation. Kohn and Allen [102] reported that 47.3% of the crude protein from soy bean meal was degraded before 16 h of incubation. Some factors that influence the hydrolysis of the protein in the rumen are the type of plant included and the forage-to-concentrate ratio in ruminant’s diet [103]. Proteases from microorganisms and feed can show additive or synergistic effects on protein degradation [103,104]. High protein hydrolysis in the rumen sometimes leads to inefficient utilization of nitrogen (N) for the synthesis of microbial protein, causing N losses (when the rate of microbial protein degradation is faster than the rate at which microorganisms incorporate ammonia into the microbial protein) [105]. Glycosylation (controlled enzymatic modification of a protein through the addition of a sugar molecule) increases the stability of EFEs, protecting enzymes against proteolytic inactivation under ruminal-like conditions [106].

## 5. Designing Fibrolytic Exogenous Enzymes (EFE)

EFEs can be obtained under conditions that act on specific substrates and environments [30]. Enzyme-protecting technology began in the decade 1960 in the pharmaceutical, textile, paper, lignocellulose hydrolysis, and bioremediation industries. Most fungal enzymes have been used in free form, but their protection might optimize them and even increase their activity. Some enzymes, such as Lac, act in the oxidation of substrates, and after releasing the product(s), their active sites are again free to act on a new substrate (recyclable); therefore, in the bioremediation, textile, and paper industries, some protection methods have been tested to improve the reutilization of enzymes [52,53]. Enzyme immobilization methods have been widely used in the last four decades because they prevent enzymes’ denaturation and proteolysis; this helps to maintain their full performance status for successful use in various industrial processes, and improves the activity, stability, and substrate specificity of enzymes [107].

Recent studies have included the protection of β-glucosidases to improve lignocellulose complex degradability [55]. Attitalla and Salleh [108], immobilized the spores of two strains of *Trichoderma harzianum* with the adjuvants xylan and carboxymethylcellulose (CMC) on alginate spheres (carboxymethylcellulases (E.C. 3.2.1.203) and xylanase were produced in vitro on free suspensions of fungal spores). Spores trapped in the alginate improved the enzyme production; additionally, alginate encapsulation prolonged the metabolic fungal activity and slowed the release of microbial spores in the media, optimizing the production of EFEs. Immobilization improves fibrolytic enzyme activity by increasing enzyme stability and improving their shelf-life [55,109]. Immobilization might allow the design and control of EFE activities under specific conditions, enabling optimal EFE activity under similar ruminal conditions.

### 5.1. Changing the Optimal pH and Temperature Condictions of Enzyme Activities

Immobilization allows the design of EFE products that act under specific pH and temperature conditions [109]. Dragomirescu et al. [19] lyophilized EFE extracts obtained from *T. viride*. The enzymes were immobilized on porous matrices using physical adsorption methods in ceramic matrices and entrapment in sol-gel glasses, and physical absorption on a ceramic support 1 to 2 times the cellulases activities. The protected EFEs enabled the hydrolysis of CMC at an optimal pH less than acid (over 5.0) and a colder temperature (20 °C less than the normal optimal temperature of EFEs).

### 5.2. Inmobilization of Enzymes to Avoid Specific Enzyme Activity Deactivation

Specific immobilization can prevent the deactivation of enzymes. Eckard et al. [110] tested colloidal proteins that formed monolayers on the hydrophobic surfaces of enzymes. The steric barrier of caseins and whey proteins adsorbed by lignocellulosic biomass increased the cellulase activities by inhibiting their adsorption in non-productive sites.

### 5.3. Changing the Specific Action Sites of Enzymes

To change the specific action sites of EFEs. Blanchette et al. [111], obtained EFEs from *T. viride* immobilized with polystyrene nanospheres, which showed higher affinity on microcrystalline cellulose (immobilized vs. non-immobilized enzymes), and were more efficient at degrading the natural cellulose structures of wood-thickened cell walls. Enzymes bound to nanoparticles might enhance the catalytic efficiency in physically insoluble substrates.

### 5.4. Designing Multi-Enzymatic Cocktails

It is possible to immobilize different combinations of ligninase, cellulase, and xylanase nanoparticles [111]. The co-immobilization can increase the efficiency of cellulose and hemicellulose degradability. By combining different activities and protection methods, it is possible to obtain sequential enzymatic reactions [6,107,112]; three simultaneously co-immobilized cysteine-labeled cellulases, including endoglucanase, exoglucanase and β-glucosidase in gold nanoparticles; and nanoparticles of magnetic silica bound in gold. These improve the release of cellobiose and glucose from cellulose. The use of various technologies associated with proteins could increase the degradation of the cellulose of raw materials for biofuel.

### 5.5. Re-Association of Enzymes and Emulation of Natural Protection Structures

It is possible to obtain artificial structures like the cellulosomes present in cellulolytic *Clostridium* and *Ruminococcus* species [113]. Complexes reconstituted from cellulases cocktails of cellulosomes allow enzymes to reattach to cellulosomes [114]. In vitro re-association of artificial enzymes with cellulosomes enable the synergic action of different enzymes on substrates compared to non-complexed components, in addition to increasing enzyme stability [114].

Although, the research on immobilization in ruminant nutrition has focused on the stabilization of exogenous enzymes using surfactants [115], and on the protection of amino acids from rumen fermentation [116], there is still an area of growing opportunity in the study of food exogenous-enzyme protection for ruminants through immobilization.

## 6. Applicability

Specific isoforms of fungal enzymes must be identified for the optimal degradation of cellulose and hemicellulose in a ruminal environment. Cocktails of enzymes can be designed or biotechnologically manipulated for the optimal catalysis or affinity of substrates of cell walls components in the rumen. Although cellulose- and hemicellulose-derived products have been widely used in the biofuel, cosmetic, paper and pharmaceutical industries, for ruminant feed applications, specific EFE products could allow the inclusion of more agricultural waste; in turn, this could reduce economic and environmental meat and milk production costs. Biotechnology can play a role in finding novel enzymatic products, but can also provide tools that allow the fusion and protection of different enzymes, which might act optimally inside the rumen at certain stages of fermentation. However, multidisciplinary efforts should be joined to obtain novel enzymatic products for ruminant feed applications. 

## 7. Conclusions

White-rot fungi produce fibrolytic and ligninolytic extracellular enzymes that degrade lignocellulosic cell wall compounds for growth. Numerous types of fibrolytic enzymes are released according to cell wall’s compounds; therefore, culture media promote specific isoforms of cellulases and xylanases. EFE products for ruminant feed applications should be obtained by culturing fungi in media that contain the fibrous forages used in diets. Designing processes to obtain and evaluate EFEs for the enhancement of NDF degradability should be performed, taking into consideration the ruminal temperature, pH, and osmolarity conditions; furthermore, EFE synergisms and interactions with ruminal microbiotic and endogenous fibrolytic enzymes that can be studied through enzyme activity evaluation and novel biotechnological tools—such as metagenomics and metatranscriptomics—should be included. Further studies might consider the cellulase-to-xylanase ratio to analyze the effect of including EFEs in ruminant diets on productive behavior, to obtain more consistent results. Additionally, technologies for protection (immobilization) allow researchers to obtain EFE products that might act at specific moments; that show more stability under ruminal pH and temperature conditions, with specific action sites; and to design multi-enzymatic cocktails that should release their different enzymatic activities at different moments. This can emulate natural fungal enzyme cocktails, to re-associate enzymes, and to emulate natural protection structures such as cellulosomes.

## Data Availability

Not applicable.

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
