# Peer review of "Improvement of Ruminal Neutral Detergent Fiber Degradability by Obtaining and Using Exogenous Fibrolytic Enzymes from White-Rot Fungi"

_animals, 2022, doi:10.3390/ani12070843_

Round 1
Reviewer 1 Report
Line 59 to 64: “Forage’s cell walls are composed of different proportions of cellulose, hemicellulose, pectin, lignin, and minerals that vary among the species and growth stage of plants [1]. Neutral detergent fiber (NDF) quantifies most cell wall’s components [2], Cellulose and hemicellulose are the most abundant proportions of cell walls but their energy availability for ruminants depend on the NDF composition and proportion [3-5], and on the efficiency of ruminal fibrolityc enzymes and bacteria to digest those components [6]”. I suggest inclusion information about the interactions between lignin and cell wall carbohydrates with reference to the covalent bonds existing between lignin and hemicellulose. Emphasize the inexistence of the enzyme ferulic acid esterase in the rumen and the importance of fungi to produce these enzymes.
Line 73 o 75: “In these cases, the removal of lignin and/or hemicellulose is the first step to expose the cellulose to the attack by cellulolytic enzymes to release glucose, due to the hydrolysis of the polysaccharide fraction NDF to fermentable sugars”. Also consider the importance of monosaccharides (pentoses, hexoses) contained in hemicellulose as a source of energy for ruminal microorganisms.
Line 149 to 150. “In the laboratory, quantification of lignin can be done with permanganate (lignin fraction of manganate or 72% H2SO4 (lignin fraction in acid detergent; ADL) [2]". Replace to: In the laboratory, quantification of lignin can be done through hydrolysis using permanganate (lignin fraction of manganate) or application of 72% H2SO4 to solubilize cellulose (lignin fraction in acid detergent; ADL).
Line 219 to 221: “Cellulase and xylanase biotechnology began in the early 80´s, by the study of the action of enzymes extracted mainly from Aspergillus sp. and Trichoderma sp. as treatments to improve the quality of food for animals [49]”. Replace to: Cellulase and xylanase biotechnology began in the early 80´s, by the study of the action of enzymes extracted mainly from Aspergillus sp. and Trichoderma sp. as treatments to improve the quality of feed for animals [49]”.
Line 295 to 296: “Optimal ruminal pH usually ranges from 6 to 7. Food fermentation in the rumen produces volatile fatty acids (VFA) and, occasionally, lactic acid”. Replace to: Optimal ruminal pH usually ranges from 6 to 7. Replace to: Feed fermentation in the rumen produces volatile fatty acids (VFA) and, occasionally, lactic acid.
Line 327 to 328: “Important proportions of protein are degraded at the beginning of the ruminal fermentation. Kohn and Allen [90] reported that 47.3% of crude protein of soy flour was 328 degraded before 16 h of incubation”. Replace to: Important proportions of protein are degraded at the beginning of the ruminal fermentation. Kohn and Allen [90] reported that 47.3% of crude protein of soyabean meal was degraded before 16 h of incubation.
The review addressed relevant aspects of the endogenous fibrolytic enzymes (EFE) utilization in ruminant nutrition. Authors should consider the effects of enzymes on the physical characteristics of the fibrous fraction on mastication and, consequently, the maintenance of ruminal pH due to salivation. The application of enzymes with a high potential for hydrolysis of the covalent bonds between lignin and hemicellulose can reduce the proportion of effective NDF (eNDF) and physically effective NDF (peNDF), which may compromise rumen pH, fiber degradability, dry matter intake and animal performance.

Author Response
I, along with my coauthors, would like to ask you to consider the revised version of the manuscript entitled “Improvement of the Ruminal Neutral Detergent Fiber Degradability by Obtaining and Using Exogenous Fibrolytic Enzymes from White-Rot Fungi (ID. Animals-1631529)” for publication in Animals MDPI. In the present letter, we are providing a point-by-point description of the corrections that we made on the revised version, and we are grateful for your suggestions that improve our manuscript quality.
Overall corrections:
Before submitting present version of the manuscript, we also have checked that:
1) Acronyms and abbreviations were correct.
2) Scientific names of fungal strains were correct.
3) Format, spaces, and spelling errors.
4) Congruency of paragraphs.
5) Grammar of sentences.
6) All references are complete and in the format in the corresponding section and numbers of cites correspond to references.
7) Author’s affiliations and addresses.
Reviewer 1.
R1. The review addressed relevant aspects of the endogenous fibrolytic enzymes (EFE) utilization in ruminant nutrition. Authors should consider the effects of enzymes on the physical characteristics of the fibrous fraction on mastication and, consequently, the maintenance of ruminal pH due to salivation. The application of enzymes with a high potential for hydrolysis of the covalent bonds between lignin and hemicellulose can reduce the proportion of effective NDF (eNDF) and physically effective NDF (peNDF), which may compromise rumen pH, fiber degradability, dry matter intake and animal performance.
Author. Many thanks for your comments and suggestions, we had addressed all of them.
- Line 59 to 64: “Forage’s cell walls are composed of different proportions of cellulose, hemicellulose, pectin, lignin, and minerals that vary among the species and growth stage of plants [1]. Neutral detergent fiber (NDF) quantifies most cell wall’s components [2], Cellulose and hemicellulose are the most abundant proportions of cell walls but their energy availability for ruminants depend on the NDF composition and proportion [3-5], and on the efficiency of ruminal fibrolityc enzymes and bacteria to digest those components [6]”. I suggest inclusion information about the interactions between lignin and cell wall carbohydrates with reference to the covalent bonds existing between lignin and hemicellulose. Emphasize the inexistence of the enzyme ferulic acid esterase in the rumen and the importance of fungi to produce these enzymes.
Line 73 o 75: “In these cases, the removal of lignin and/or hemicellulose is the first step to expose the cellulose to the attack by cellulolytic enzymes to release glucose, due to the hydrolysis of the polysaccharide fraction NDF to fermentable sugars”. Also consider the importance of monosaccharides (pentoses, hexoses) contained in hemicellulose as a source of energy for ruminal microorganisms.
Author. We analyzed and addressed both suggestions, then, we included several more information about the structure and composition of hemicellulose, the synthesis of the lignin, the types of bonds between lignin’s subunits, and the linkages among lignin and hemicellulose. Additionally, we included information about the fungal enzymes that act un lignin-hemicellulose linkages.
At final we included next references:
- Grabber, J.H.; Ralph, J.; Lapierre, C.; Barriere, Y. Genetic and molecular basis of grass cell-wall degradability. I Lignin-cell wall matrix interactions. R. Biologies 2004, 327, 455-465. https://doi:10.1016/j.crvi.2004.02.009
- Nishimura, H.; Kamiya, A.; Nagata, T.; Katarhira, M.; Watanabe, T. Direct evidence for α ether linkage between lignin and carbohydrates in wood cell walls. Scientific reports 2018, 8, 6538. https://doi: 10.1038/s41598-018-24328-9
- Lu, Y.; He, Q.; Fan, G.; Cheng, Q.; Song, G. Extraction and modification of hemicellulose from lignocellulose biomass: A review. Green Processing and Synthesis 2021, 10, 779-804. https://doi.org/10.1515/gps-2021-0065
- Jung, H.D.; Casler, M.D. Maize stem tissues: Impact of development on cell wall degradability. Crop Science 2006, 46, 1801-1809. https://doi:10.2135/cropsci2006.02-0086
- Huang, L.Z.; Ma, M.G.; Ji, X.X.; Choi, S.E.; Si, C. Recent developments and applications of hemicellulose from wheat straw: A review. Frontiers in Bioengineering and Biotechnology 2021, 9, 690773. https://doi: 10.3389/fbioe.2021.690773
- Jung, H.J. Maize stem tissues: Ferulate deposition in developing internode cell walls. Phytochemistry 2003, 63(5), 543-549. https://doi.org/10.1016/S0031-9422(03)00221-8
- Tarasov, D.; Leitch, M; Fatehi, P. Lignin-carbohydrate complexes: properties, applications, analyses, and methods of extraction: a review. Biotechnology for Biofuels 2018, 11, 269. https://doi.org/10.1186/s13068-018-1262-1
- Mkabayi, L.; Malgas, S.; Wilhemi, B.S.; Pletschke, B.I. Evaluating ferulolyl esterase-xylanase synegirms for hydrocynnamic acid and xylo-olisaccharide production from untreated, hydrothermally pre-treated and dilute-acid pre-treated corn cobs. Agronomy 2020, 10, 688. https://doi:10.3390/agronomy10050688
- Van Dyk, J.S.; Pletschke, B.I. A review of lignocellulose bioconversion using enzymatic hydrolysis and synergistic cooperation between enzymes—Factors a_ecting enzymes, conversion and synergy. Biotechnology Advances 2012, 30, 1458–1480. https://doi: 10.1016/j.biotechadv.2012.03.002
- Li, X.; Griffin, K.; Langeveld, S.; Frommhagen, M.; Underlin, E.N.; Kabel, M.A.; De Vries, R.P.; Dilokpimol, A. Functional validation of two fungal subfamilies in carbohydrate esterase family 1 by biochemical characterization of esterases from uncharacterized branched. Frontiers in Bioengineering and Biotechnology 2020, 8, 694. https://doi: 10.3389/fbioe.2020.00694
- Dilokpimol, A.; Mäkelä, M.R.; Aguilar-Pontes, M.; Benoit-Gelber, I.; Hildén.; De Vries, R.P. Diversity of fungal feruloyl esterases: updated phylogenetic classification, properties, and industrial applications. Biotechnology for Biofuels 2016, 9, 231. https://doi:10.1186/s13068-016-0651-6
- Miranda-Romero, L.A.; Tirado-González, D.N.; Tirado-Estrada, G.; Améndola-Massioti, R.; Sandoval-González, L.; Ramírez-Valverde, R.; Salem, A.Z.M. Quantifying non-fibrous carbohydrates, acid detergent fiber and cellulose of forage through an in vitro gas production technique. Journal of the Science of Food and Agriculture 2020, 100(7), 3099-3110. https://doi: 10.1002/jsfa.10342
However, instead of including the revision in the Introduction section, we reorganized the second section “2. Natural production of multi-enzyme cocktails by white-rot fungi” where that information better-fit.
- Line 149 to 150. “In the laboratory, quantification of lignin can be done with permanganate (lignin fraction of manganate or 72% H2SO4 (lignin fraction in acid detergent; ADL) [2]". Replace to: In the laboratory, quantification of lignin can be done through hydrolysis using permanganate (lignin fraction of manganate) or application of 72% H2SO4 to solubilize cellulose (lignin fraction in acid detergent; ADL).
Author. Thank you, we have literally addressed the suggestion.
- Line 219 to 221: “Cellulase and xylanase biotechnology began in the early 80´s, by the study of the action of enzymes extracted mainly from Aspergillus sp. and Trichoderma as treatments to improve the quality of food for animals [49]”. Replace to: Cellulase and xylanase biotechnology began in the early 80´s, by the study of the action of enzymes extracted mainly from Aspergillus sp. and Trichoderma sp. as treatments to improve the quality of feed for animals [49]”.
Author. Thank you, we have literally addressed the suggestion.
- Line 295 to 296: “Optimal ruminal pH usually ranges from 6 to 7. Food fermentation in the rumen produces volatile fatty acids (VFA) and, occasionally, lactic acid”. Replace to: Optimal ruminal pH usually ranges from 6 to 7. Replace to: Feed fermentation in the rumen produces volatile fatty acids (VFA) and, occasionally, lactic acid.
Author. Thank you, we have literally addressed the suggestion.
- Line 327 to 328: “Important proportions of protein are degraded at the beginning of the ruminal fermentation. Kohn and Allen [90] reported that 47.3% of crude protein of soy flour was 328 degraded before 16 h of incubation”. Replace to: Important proportions of protein are degraded at the beginning of the ruminal fermentation. Kohn and Allen [90] reported that 47.3% of crude protein of soyabean meal was degraded before 16 h of incubation.
Author. Thank you, we have literally addressed the suggestion.
Again, we are grateful, we are sure that this version has been importantly improved because of your kind comments and suggestions. We hope to hear from you soon.
Sincerely,
Deli Nazmín Tirado-González, Ph. D.
Gustavo Tirado-Estrada, Ph. D.
Corresponding authors

Reviewer 2 Report
Overall Comments:
This manuscript is a review regarding the used of fungal enzymes to increase NDF degradability in ruminant species. The review is well organized and complete. My major comments are that a detail description of the search and selection criteria of the cited literature should be included and that a section describing the field applicability of this technology also should be included to improve the value of this document. Furthermore, grammar should be reviewed in the whole document. For more details, please see below specific comments.
Specific Comments:
Line 30. Please remove “the” before “cell wall”.
Line 32. Please add a comma after “(EFE)”.
Line 34. Here and elsewhere, please consider changing “productive behavior” to “productivity”.
Line 41-42. Please describe in the manuscript in detail the search criteria and selection process that was used to identify these articles.
Line 51. Please add “The” before “Present review”.
Line 55. Please remove bolt of the “w” of the word “white-rot”.
Line 61. Change comma for a period or change capitol C on the word “Cellulose”.
Line 61-63. This sentence is confusing. Cellulose and hemicellulose are included in the NDF composition. Please re-word to clarify authors’ statement.
Line 65. Please add comma after “areas”.
Line 65-70. Run-on sentence. Please divide this sentence in two or three sentences.
Line 71-76. Same comments as above.
Line 95-96. It looks that there is some missing wording in this sentence. Please revise.
Line 97. Please specify that authors are referring to withing the digestive trat of ruminant animals.
Line 100-106. Run-on sentence. Please see comments above.
Line 118-119. Please correct the hemicellulose split at the end of line 118 (hemi-).
Line 142. Authors have already used this acronym in line 121. I suggest using this abbreviation for the concept explained in Line 142.
Line 149. Please revise this sentence.
Line 155. Please add “type” after “not condensed”.
Line 156. Here are elsewhere, please don’t start a sentence with an acronym.
Line 203. The authors are listing two different categories (plant type vs plant cell composition analyses). Please reword this sentence for clarification.
Line 233. I would recommend removing or rewording for clarity the “or study (on vivo, in situ, in vivo)” phrase. Also, “in vivo” is repeated.
Line 238. Please explain a little more in detail why in vitro experiments’ results don’t directly translate to in vivo experiments.
Line 239-240. Please check grammar.
Line 242. Please verify if this is accurate. In the dairy industry, after cows start lactation ADG (average daily gain) is usually not further monitored.
Line 246-247. Please revise grammar.
Line 271. What do authors mean by “improvement of the activity NDFD”? Is this improvement of NDFD? If that the case, please remove the word “activity”. Also, the relationship is with the concentration of certain types of cellulases or their activity? Please re-word for clarity.
Line 288. Please change comma for a period before “biotechnology”.
Line 290-291. It looks that there is a second part missing for this sentence. Please revise.
Line 294. Please add the “ruminal” before “pH”.
Line 311. Same comment as above.
Line 290-317. Is the acronym AGV (Acidos Grasos Volatiles?) referring to Volatile Fatty Acids (VFA)? Please change the acronym or explain what this acronym means.
Line 330. Please add “s” after “ruminant’ “
Line 339. Please reword/modify this subtitle. It is confusing what authors are referring to.
Line 345. Please revise grammar/sentence structure.
Line 358. Please remove extra “the” before “fungal activity”.
Line 402. Adding a section regarding the field applicability would greatly increase the value of this document.
Author Response
I, along with my coauthors, would like to ask you to consider the revised version of the manuscript entitled “Improvement of the Ruminal Neutral Detergent Fiber Degradability by Obtaining and Using Exogenous Fibrolytic Enzymes from White-Rot Fungi (ID. Animals-1631529)” for publication in Animals MDPI. In the present letter, we are providing a point-by-point description of the corrections that we made on the revised version, and we are grateful your kind suggestions that improve our manuscript quality.
Overall corrections:
Before submitting present version of the manuscript, we also have checked that:
1) Acronyms and abbreviations were correct.
2) Scientific names of fungal strains were correct.
3) Format, spaces, and spelling errors.
4) Congruency of paragraphs.
5) Grammar of sentences.
6) All references are complete and in the format in the corresponding section and numbers of cites correspond to references.
7) Author’s affiliations and addresses.
Reviewer 2.
Author. We are grateful with your revision. We have made all the punctual corrections and addressed the grammar revision. Additionally, to follow the suggestions of Reviewer 1, we have to make several changes in the second section “2. Natural production of multi-enzyme cocktails by white-rot fungi” to include several more information about the structure and composition of hemicellulose, the synthesis of the lignin, the types of bonds between lignin’s subunits, and the linkages among lignin and hemicellulose, and about the fungal enzymes that act un lignin-hemicellulose linkages. Then, we had to restructure and reorganized information of that section.
- Line 30. Please remove “the” before “cell wall”.
Author. Done.
- Line 32. Please add a comma after “(EFE)”.
Author. Done.
- Line 34. Here and elsewhere, please consider changing “productive behavior” to “productivity”.
Author. Done.
- Line 51. Please add “The” before “Present review”.
Author. Done.
- Line 55. Please remove bolt of the “w” of the word “white-rot”.
Author. Done.
- Line 61. Change comma for a period or change capitol C on the word “Cellulose”.
Author. Done.
- Line 61-63. This sentence is confusing. Cellulose and hemicellulose are included in the NDF composition. Please re-word to clarify authors’ statement.
Author. Done.
- Line 65. Please add comma after “areas”.
Author. Done.
- Line 65-70. Run-on sentence. Please divide this sentence in two or three sentences.
Author. Done.
- Line 71-76. Same comments as above.
Author. Done.
- Line 95-96. It looks that there is some missing wording in this sentence. Please revise.
Author. Done.
- Line 97. Please specify that authors are referring to withing the digestive trat of ruminant animals.
Author. Thank you, we addressed the suggestion, and in the corresponding lines we were more specific about the kind of experiments and results reported about the how fungi require the coordinated action of a cocktail of enzymes. In addition, we included the description about the experiments performed in ruminal liquid.
Lines 263-277: “As in nature, efficient enzyme cocktails of enzymes are required to complete hydrolysis of lignocellulosic biomass. For example, increasing the ratio of endoglucanases: xylanases could improve the NDF degradability of alfalfa and corn silages [57]. Fortes et al. [58] tested fungal enzyme extracts on the degradation of the lignocellulosic complex sugarcane: Cellulases and xylanases of Trichoderma reesei and Aspergillus awamori were acid ferulic esterases (1700 vs. 420 UI/L), endoglucanases (CMCase; 20,000 vs. 4,900 UI/L), β-glucosidase (340 vs. 45,600 UIL/), and xylanases (12,600 vs. 79,000 UI/L), respectively, and found that enzymatic activity of T. reesei and A. awamori enzyme mixtures acted synergistically by 2-fold. Mkabayi et al. [32], tested the synergistic action of two feruloyl esterases (FAE5 and FAE6), and an endo-xylanase (Xyn11) from Thermomyces lanuginosus, were applied to 1% (w/v) corn cobs. The combination of 66% Xyn11 and 33% FAE6 displayed an improvement in reducing sugars compared to Xyn11 alone. Xylanases generate ferulated xylo-oligosaccharides (XOS), which become substrates for FAE, liberating ferulic acid as a product. The removal of ferulic acid then increases the accessibility of xylanases to XOS, for further hydrolysis into shorter XOS and/or xylose”.
Lines 101-109: “To access to cellulose and hemicellulose energy availability, in nature, fungi require the coordinated action of ligninases, cellulases, and xylanases, acting synergistically [26, 27, 30], therefore, lignocellulose hydrolysis also requires the coordinated action of cocktails of enzymes [31, 32]. Tirado-González et al. [30], found that NDF, acid detergent fiber (ADF), and ADL culture medias extracted from different corn stover hybrids, promote different production of exogenous cellulolytic and xylanolytic enzymes ratios of Bjerkandera adusta, Sporotrichum pulverulentum, Trametes trogii, and P. ostreatus, thus, culture media can induce specific forms and isoforms of enzymes to useful to degrade cell walls of high-fibrous forages included in ruminant nutrition”.
- Line 100-106. Run-on sentence. Please see comments above.
Author. Done.
- Line 118-119. Please correct the hemicellulose split at the end of line 118 (hemi-).
Author. Done.
- Line 142. Authors have already used this acronym in line 121. I suggest using this abbreviation for the concept explained in Line 142.
Author. Done.
- Line 149. Please revise this sentence.
Author. Done.
- Line 155. Please add “type” after “not condensed”.
Author. Done.
- Line 156. Here are elsewhere, please don’t start a sentence with an acronym.
Author. Done.
- Line 203. The authors are listing two different categories (plant type vs plant cell composition analyses). Please reword this sentence for clarification.
Author. Done.
- Line 233. I would recommend removing or rewording for clarity the “or study (on vivo, in situ, in vivo)” phrase. Also, “in vivo” is repeated.
Author. Done.
- Line 239-240. Please check grammar.
Author. Done.
- Line 246-247. Please revise grammar.
Author. Done.
- Line 271. What do authors mean by “improvement of the activity NDFD”? Is this improvement of NDFD? If that the case, please remove the word “activity”. Also, the relationship is with the concentration of certain types of cellulases or their activity? Please re-word for clarity.
Author. Done. In fact, it was incorrectly redacted We had corrected the sentence.
- Line 288. Please change comma for a period before “biotechnology”.
Author. Done.
- Line 290-291. It looks that there is a second part missing for this sentence. Please revise.
Author. Done.
- Line 294. Please add the “ruminal” before “pH”.
Author. We have added “the” before “ruminal pH”.
- Line 311. Same comment as above.
Author. We have added “the” before “ruminal pH”.
- Line 290-317. Is the acronym AGV (Acidos Grasos Volatiles?) referring to Volatile Fatty Acids (VFA)? Please change the acronym or explain what this acronym means.
Author. The acronym was incorrect, we had changed it by “VFA (Volatile Fatty Acids instead of Ácidos Grasos Volátiles).
- Line 330. Please add “s” after “ruminant”
Author. Done.
- Line 339. Please reword/modify this subtitle. It is confusing what authors are referring to.
Author. We had changed the subtitle “Enzyme protection designed as an alternative increase their activity in ruminal conditions” for a shorter one, to make it clearer (“Designing exogenous fibrolytic enzymes (EFE)”).
- Line 345. Please revise grammar/sentence structure.
Author. Done.
- Line 358. Please remove extra “the” before “fungal activity”.
Author. Done.
- Line 41-42. Please describe in the manuscript in detail the search criteria and selection process that was used to identify these articles.
Author. Done.
- Line 238. Please explain a little more in detail why in vitro experiments’ results don’t directly translate to in vivo
We have also rewritten those sentences to clarify the original mean intention of authors cited in references. After reviewing the articles cited in that section, we realized that no one related in vitro and in vivo EFE responses. Therefore, we changed “sometimes in vitro results are not suitable with in vivo effects [7, 27, 54, 55]. Including EFE in ruminant diets can have more effects in vivo but they might vary are more than in vitro studies [56, 57]” for “In vitro results of EFE addition are not always suitable with in vivo effects, but sometimes they are comparable, for example, Eun et al. [57] found that some doses of endoglucanases and xylanases could enhance the IVNDFD that is accord with in vivo studies that show the increasing of NDFD when EFE are added to ruminant diets [67]. Additionally, the in vitro studies of the effect of EFE addition in ruminant diets allow to understand the interactions among the type of activities of EFEs and the ruminal liquid environment, with more consistent results than in vivo studies [57, 68, 69] (Lines 304-310).
- Line 242. Please verify if this is accurate. In the dairy industry, after cows start lactation ADG (average daily gain) is usually not further monitored.
Author. Neither Gaddo et al. (2009) nor Arriola et al. (2011) address their individual studies in different stages of lactation, however Arriola et al. (2011) and Gado et al. (2009), studied the effect of EFE at a different stage of lactation and reported similar results. Although, we understand that due to the environmental differences those results might not been compared among them, we considered interesting to present both to lectors.
- Line 402. Adding a section regarding the field applicability would greatly increase the value of this document.
Author. Some of the possible applications of the information are included in the main text, then, we emphasized them in a section under the subtitle “5. Applicability”.
Again, we are grateful, we are sure that this version has been importantly improved because of your revision. We hope to hear from you soon.
Deli Nazmín Tirado-González, Ph. D.
Gustavo Tirado-Estrada, Ph. D.
Corresponding authors

Reviewer 3 Report
I have no comment on the content.
Some specific comments are as follows,
L141; degradability of the NDF (NDFD). --- NDFD.
L164; (acid detergent lignin (ADL)) --- ADL
L169; copper ions (Cu) --- Cu2+ ?
L186; MnP --- Coming out for the first time ?
L203; acid detergent fiber (ADF), and acid detergent lignin (ADL) --- ADF, and ADL
L208; Obodai, Cleland-Okine, and Vowotor [44] --- Obodai et al. [44]
L227; 3) reduce feed conversion --- increase ?
L231; IVDNDFD --- IVNDFD ?
L236; IVDNDF --- IVNDFD ?
L268; ratios from 0.01 --- You mean ratio 1:100 ?
L290; AGV --- Coming out for the first time ?
L301; may negative affect --- may negatively affect ?
L312; mosmol --- milliosmole ? mOsm ?
L356; in vitro --- Italic
Author Response
I, along with my coauthors, would like to ask you to consider the revised version of the manuscript entitled “Improvement of the Ruminal Neutral Detergent Fiber Degradability by Obtaining and Using Exogenous Fibrolytic Enzymes from White-Rot Fungi (ID. Animals-1631529)” for publication in Animals MDPI. In the present letter, we are providing a point-by-point description of the corrections that we made on the revised version, and we are grateful for your suggestions that improve our manuscript quality.
Overall corrections:
Before submitting present version of the manuscript, we also have checked that:
1) Acronyms and abbreviations were correct.
2) Scientific names of fungal strains were correct.
3) Format, spaces, and spelling errors.
4) Congruency of paragraphs.
5) Grammar of sentences.
6) All references are complete and in the format in the corresponding section and numbers of cites correspond to references.
7) Author’s affiliations and addresses.
Reviewer 3.
Author. We are grateful with your revision. We have made all the punctual corrections and addressed the grammar revision. Additionally, to follow the suggestions of Reviewer 1, we have to make several changes in the second section “2. Natural production of multi-enzyme cocktails by white-rot fungi” to include several more information about the structure and composition of hemicellulose, the synthesis of the lignin, the types of bonds between lignin’s subunits, and the linkages among lignin and hemicellulose, and about the fungal enzymes that act un lignin-hemicellulose linkages. Then, we had to restructure and reorganized information of that section.
- L141; degradability of the NDF (NDFD). --- NDFD.
Author. Thank for the comment, we widely discussed about using NDFD for degradability of the NDF, the problem is to differentiate “NDF degradability” and “NDF digestibility”, we considered NDF degradability that is used to overall describe results of studies performed in vitro, in situ or in vivo meanwhile NDF digestibility is evaluated exclusively in in vivo experiments. Then, to differentiate all of them, we decided to use NDF degradability for overall results (whether in vitro, in situ or in vivo), NDFD for NDF in vivo digestibility, ISNDFD for in situ NDF disappearance, and IVNDFD for in vitro NDF degradability.
- L164; (acid detergent lignin (ADL)) --- ADL
Author. Corrected.
- L169; copper ions (Cu) --- Cu2+ ?
Author. Thank you, you are correct. We have changed it.
- L186; MnP --- Coming out for the first time?
Author. There was an error in the redaction of the statement, we had changed the statement “MnP peroxidase source” for “manganese peroxidases (MnP) to increase”. The abbreviature has been described at the beginning of the previous paragraph.
- L203; acid detergent fiber (ADF), and acid detergent lignin (ADL) --- ADF, and ADL
Author. Correct, acid detergent lignin and acid detergent fiber were previously defined, therefore we have made the correction in this line, however, in line 203 is the first time that acid detergent fiber has been mentioned.
- L208; Obodai, Cleland-Okine, and Vowotor [44] --- Obodai et al. [44].
Author. Corrected.
- L227; 3) reduce feed conversion --- increase?
Author. Yes, since FC = DMI (dry matter intake)/body weight gain or milk production. Then, when EFE are added in ruminant feedstuff overall digestibility can improve and ruminants could decrease the DMI without negative effects on meat or milk production.
- L231; IVDNDFD --- IVNDFD?
Author. Corrected.
- L236; IVDNDF --- IVNDFD?
Author. Corrected.
- L268; ratios from 0.01 --- You mean ratio 1:100?
Author. That is right, therefore, we changed the ratio 0.01:1 for 1:100.
- L290; AGV --- Coming out for the first time?
Author. Corrected. There was an error, the correct acronym is VFA (volatile fatty acids), we had changed it by “VFA (Volatile Fatty Acids instead of Ácidos Grasos Volátiles (AGV)). In addition, the acronym is coming out the first time in this line, therefore we had corrected and defined it in this line and omitted the description in subsequent lines.
- L301; may negative affect --- may negatively affect?
Author. Corrected. We have changed the statement “may have negative effects on” instead of “may negative affect”.
- L312; mosmol --- milliosmole? mOsm?
Author. That is right, we had changed the unity.
- L356; in vitro --- Italic
Author. Corrected.
Again, we are grateful, we are sure that this version has been importantly improved because of your kind comments and suggestions. We hope to hear from you soon.
Sincerely,
Deli Nazmín Tirado-González, Ph. D.
Gustavo Tirado-Estrada, Ph. D.
Corresponding authors
